# Safety Aware Constrained Reinforcement Learning via Contrastive State Representations

## Abstract

In model-free Safe Reinforcement Learning (Safe RL) methods, agents are tasked with satisfying constraints in high-dimensional environments. However, they often learn from state representations that do not explicitly encode safe or unsafe information. This forces them into a prolonged trial-and-error cycle where the agent's learning process is split between constraint satisfaction and maximizing rewards. We argue that this is not fundamentally a policy learning problem, but a representation problem. To address this, we introduce a framework - Self Supervised Safe Reinforcement Learning (S3RL) - that jointly learns a control policy and safety-aware state representations. These representations are learned by maximizing the mutual information (MI) between state embeddings and their corresponding safety labels. We optimize the MI objective using a contrastive InfoNCE loss, which learns to distinguish safe states from unsafe ones. Our representation learning module is algorithm agnostic and can be integrated into various Safe RL algorithms. Integrating it into a Lagrangian-based soft actor-critic update, we prove that our joint objective guarantees stable and monotonic policy improvement. Experiments on multiple safety environment benchmarks validate that our method helps in alleviating the conflict between exploration and constraint satisfaction, leading to policies that achieve higher rewards than state-of-the-art Safe RL baselines without compromising safety.

## 1 Introduction

A major challenge in Reinforcement Learning (RL) is making sure that agents are safe while maximizing the cumulative rewards. The field of Safe RL addresses this by framing the problem as a constrained sequential decision-making task, where an agent must maximize its reward while satisfying safety constraints, which are typically enforced by limiting the expected cumulative value of cost functions (Gu et al., 2022). Modern approaches solve these problems by training deep neural networks from scratch, learning from repeated interactions with an environment. To enforce safety, these methods typically use penalization mechanisms, such as Lagrangian multipliers, which guide the agent towards constraint-satisfying regions (Achiam et al., 2017; Ray et al., 2019; Stooke et al., 2020; Liu et al., 2022).

While effective, these methods have a major limitation: the agent learns to be safe through extensive interactions, and does not not develop an explicit awareness of safe regions, but rather learn to avoid certain actions purely based on penalties from cost functions. This means that the agent's learning capacity is split between two tasks: (1) understanding which parts of the environment incur cost violations and (2) maximizing reward, which forces the policy to learn about safety through repeated, costly violations, which is sample-inefficient and can compromise the agent's ability to focus on reward maximization.

We argue that this is not fundamentally a policy learning problem, but a representation problem. The main challenge is not just reacting to a violation, but knowing how to avoid unsafe regions in the first place. Therefore, we propose a different approach: to decouple the task of learning about safety from the task of policy optimization. We posit that a better and more efficient way is first to understand a state representation that already encodes information about the safe state space (Figure 1), and train a

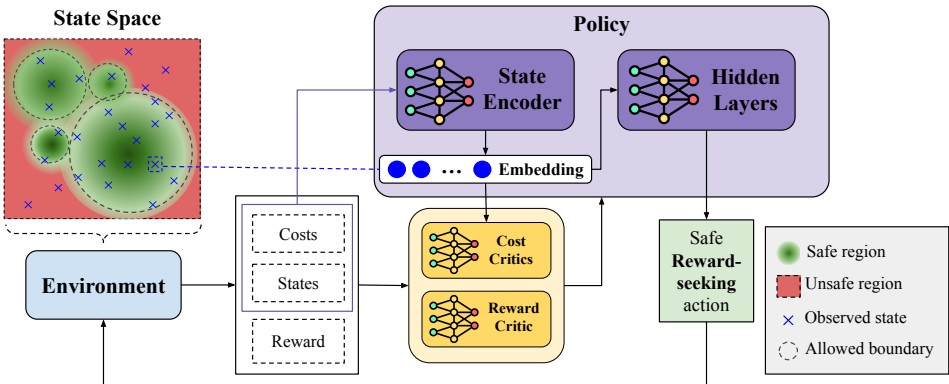

Figure 1: Overview of the safety aware reinforcement learning framework (S3RL). The agent (policy) interacts with the environment by sending actions and receiving states, costs and rewards. States are encoded into a latent embedding space via a state encoder. In the embedding space, regions corresponding to safe behaviors (green) and unsafe behaviors (red) are identified. A safety feedback loop provides corrective signals from unsafe embeddings back to the policy, allowing the agent to adjust its behavior under safety constraints while maximizing the rewards.

policy using the learned representation. By giving the agent a better map of the environment's risks, the policy learner can be better positioned to focus on maximizing rewards.

While prior works (Laskin et al., 2020; Eysenbach et al., 2022) have successfully applied contrastive learning in RL, their objectives remain largely *task-agnostic*. In these methods, positives and negatives are constructed from either temporal proximity, future prediction, or data augmentation. Although such approaches improve sample efficiency by enriching representations, they also do not explicitly address the challenge of safety in RL. This leaves a critical gap: existing contrastive methods provide general-purpose feature learning but fail to encode the boundaries that determine safe versus unsafe behaviors. Our work targets this gap by leveraging *safety labels as semantically meaningful supervision*, where positives correspond to safe states and negatives to unsafe states. By shaping the latent space according to safety labels and embedding safety-awareness directly into the policy optimization process, our approach ensures that learned representations capture the boundaries critical for constraint satisfaction. In doing so, the contribution reframes representation learning as a *safety-critical inductive bias* that alleviates the costly trial-and-error process typically required by conventional model-free Safe RL methods.

To this end, we introduce a modular, self-supervised method for learning these safety-aware state representations. We maximize a mutual-information objective via an InfoNCE loss built from safe/unsafe labels to train a state encoder that explicitly encodes proximity to unsafe regions. Our approach is algorithm-agnostic and can be integrated with both modern Safe RL and standard RL methods. Our contributions are:

1. We identify and analyze a state-representation bottleneck in Safe RL: standard pipelines lack explicit safety awareness in the state space, forcing trial-and-error learning that splits capacity between constraint satisfaction and reward maximization.

2. We introduce a self-supervised module that learns safety-aware representations by maximizing the mutual information between policy embeddings and per-transition safety labels using an InfoNCE objective constructed from safe/unsafe replay partitions.

3. We propose a decoupled training scheme that applies the MI regularizer only to the policy encoder while keeping the critic encoder separate, and we outline online/semi-online synchronization schedules that mitigate non-stationarity and preserve off-policy stability.

4. We provide theoretical guarantees: a per-state soft policy-improvement lemma and a block-coordinate ascent result showing monotone increase of a joint objective combining the soft Lagrangian return with the MI term, ensuring stable improvement under safety shaping.

5. We demonstrate algorithm-agnostic integration and consistent reward improvements on safety control tasks while maintaining safety constraints within acceptable limits.

## 2 PRELIMINARIES

**Constrained Markov Decision Process.** Safe RL problems are typically formulated as Constrained Markov Decision Processes (CMDPs) (Altman, 1999). A CMDP is defined by the tuple $(\mathcal{S}, \mathcal{A}, P, r, \mathcal{C}, \mu_0, \gamma)$, where $\mathcal{S}$ is the state space, $\mathcal{A}$ is the action space, $P : \mathcal{S} \times \mathcal{A} \times \mathcal{S} \to [0, 1]$ is the transition probability function, $r : \mathcal{S} \times \mathcal{A} \to \mathbb{R}$ is the reward function, $\mu_0$ is the initial state distribution, and $\gamma \in [0, 1)$ is the discount factor. Safety violations are incorporated through a set of $m$ cost functions, $\mathcal{C} = \{c_i : \mathcal{S} \times \mathcal{A} \to \mathbb{R}_{\geq 0}\}_{i=1}^m$. Each cost function quantifies an undesirable behavior (e.g., entering unsafe regions or exceeding operational limits) and is associated with a safety threshold $\epsilon_i$.

An agent interacts with the environment by executing a policy $\pi(a_t|s_t)$, generating trajectories $\tau = (s_0, a_0, s_1, \dots)$ where the initial state $s_0$ is sampled from $\mu_0$. In the discounted setting, a policy $\pi$ is evaluated based on the expected discounted return for the reward, $J_r(\pi) = \mathbb{E}_{\tau \sim \pi}\left[\sum_{t=0}^{\infty} \gamma^t r(s_t, a_t)\right]$, and similarly for each constraint, we have the expected discounted returns per cost function $J_{c_i}(\pi) = \mathbb{E}_{\tau \sim \pi}\left[\sum_{t=0}^{\infty} \gamma^t c_i(s_t, a_t)\right]$, $\forall i \in \{1, \dots, m\}$. The objective in a CMDP is to find an optimal policy $\pi^*$ that maximizes the expected reward return while ensuring that each expected cost return remains below its respective threshold:

$$\max_{\pi} \quad J_r(\pi) := \int d^\pi(s, a) \left[r(s, a) + \alpha \, \mathcal{H}(\pi(\cdot|s))\right] da \, ds$$

$$\text{s.t.} \quad J_{c_i}(\pi) := \int d^\pi(s, a) \, c_i(s, a) \, da \, ds \leq \epsilon_i, \quad i = 1, \dots, m, \tag{1}$$

where $d^\pi(z, a)$ is the induced occupancy and $\mathcal{H}(\pi(\cdot|s))$ represents the entropy of the policy $\pi$. The occupancy $d^\pi(z, a)$ is to capture both task-relevant and safety-relevant features by leveraging policy state encoder $f_{\phi_\pi}$, which maps raw state $s$ into latent embedding $z = f_{\phi_\pi(s)}$. Note that the entropy term $\mathcal{H}(\pi(\cdot|s))$ is only present in soft policy optimization methods, such as Soft Actor-Critic (SAC) (Haarnoja et al., 2018). The coefficient $\alpha$ controls the weight of this entropy regularization term.

## 3 METHODOLOGY

We hypothesize that an effective encoder maps raw observations into a latent representation where safety-relevant structure — such as proximity to hazards or constraint boundaries — is linearly separable and predictive of future violations. A well-structured embedding serves a critical role in policy learning. It *conditions the policy* on features that align action logits with safe state–action regions, thereby improving the entropy-regularized objective while simultaneously respecting constraints. Thus, it improves the agent's ability to organize its representation space in a way that accelerates reward maximization. To achieve this, we perform mutual information maximization between safety labels and latent embeddings, explicitly encoding safety structure in the learned representation space.

### 3.1 MUTUAL INFORMATION AS A SAFETY-AWARE REPRESENTATION OBJECTIVE

For state space $S$, let $Z = f_{\phi_\pi}(S)$ denote the policy's latent representation and $L \in \{0, 1\}$ the safety label indicating whether a transition is safe or unsafe. The mutual information $I(Z; L)$ measures how much knowing the embedding $Z$ reduces uncertainty about $L$. For two random variables $Z$ and $L$ with joint density $p(Z, L)$ and marginals $p(Z)$ and $p(L)$, the mutual information is defined as:

$$I(Z; L) \;=\; \mathbb{E}_{p(Z,L)}\left[\log \frac{p(Z, L)}{p(Z)\, p(L)}\right] \;=\; \iint p(z, l) \log \frac{p(z, l)}{p(z)\, p(l)} \, dz \, dl. \tag{2}$$

This is equivalent to the Kullback–Leibler divergence between the joint distribution and the product of its marginals:

$$I(Z; L) = D_{\mathrm{KL}}(p(Z, L) \,\|\, p(Z)\, p(L)) = \mathcal{H}(L) - \mathcal{H}(L|Z) \tag{3}$$

Maximizing this quantity encourages $f_{\phi_\pi}$ to retain and organize safety-relevant information in its latent space i.e. that observing $Z$ reduces uncertainty about $L$. Intuitively, large $I(Z; L)$ forces $f_{\phi_\pi}$ to encode safety-discriminative structure, sharpening the shaped policy gradients that depend on the critic estimate $\widetilde{Q}_\lambda$. Such safety-aware geometry makes policy gradients more aligned with constraint satisfaction, especially in off-policy updates where replay data contain both safe and unsafe transitions.

### 3.2 POLICY IMPROVEMENT WITH A DECOUPLED MI REGULARIZER

We fix the critics and dual variables, and define the shaped value as $\widetilde{Q}_\lambda(s, a) = Q_r(s, a) - \lambda^\top Q_c(s, a)$, where $Q_r(s, a)$ is the reward critic and $Q_c(s, a)$ is the cost critic. These values are computed on features from the *critic* encoder and are therefore independent of the policy encoder. We define the *soft Lagrangian return* for a policy $\pi$:

$$\mathcal{J}_{\text{soft}}(\pi; \lambda) := \int_{\mathcal{S}} d^\mu(s) \underbrace{\int_{\mathcal{A}} \pi(a|s) \left[ \widetilde{Q}_\lambda(s, a) - \alpha \log \pi(a|s) \right] da}_{:= \Phi_\lambda(\pi; s)} ds, \qquad (4)$$

for any reference state distribution $d^\mu$ (e.g., replay marginal). Here, $\Phi_\lambda(\pi; s)$ denotes the *per-state soft objective*, i.e., the expected shaped return obtained by following policy $\pi$ at state $s$ under the Lagrangian-augmented value $\tilde{Q}_\lambda(s, a)$. Intuitively, this quantity measures how well the current policy balances reward maximization, safety penalties, and entropy regularization at a given state. Importantly, the effectiveness of this objective depends on how $s$ is represented in the policy encoder (Figure 1). If the encoder fails to capture features that distinguish safe from unsafe regions, then $\Phi_\lambda(\pi; s)$ may not reliably reflect the safety consequences of actions. This creates a representation bottleneck: the policy update improves $\Phi_\lambda$ only with respect to the features currently available, which may be insufficient for safety-critical decisions. Given the policy encoder $f_{\phi_\pi}$ which maps raw observations to latent representations, $Z = f_{\phi_\pi}(S)$ and safety label $L$, we augment the objective in Equation 4 with a mutual-information term:

$$\mathcal{F}(\pi, \phi_P; \lambda) := \mathcal{J}_{\text{soft}}(\pi; \lambda) + \beta \, I(Z; L), \qquad \beta \geq 0. \qquad (5)$$

**Lemma 1** (Per-state soft improvement). *For any fixed $s$ and $\lambda$, define the Boltzmann policy $\pi^*(a|s) \propto \exp\left( \frac{1}{\alpha} \widetilde{Q}_\lambda(s, a) \right)$. Then*

$$\Phi_\lambda(\pi^*; s) - \Phi_\lambda(\pi; s) = \alpha \, \text{KL}\big( \pi(\cdot|s) \, \| \, \pi^*(\cdot|s) \big) \geq 0, \qquad (6)$$

*with equality iff $\pi(\cdot|s) = \pi^*(\cdot|s)$ a.e.*

*Proof.* Rewrite $\Phi_\lambda(\pi; s)$ as $\Phi_\lambda(\pi; s) = -\alpha \, \text{KL}(\pi(\cdot|s) \, \| \, \pi^*(\cdot|s)) + \alpha \log Z_\lambda(s)$, where $Z_\lambda(s) = \int \exp\left( \widetilde{Q}_\lambda(s, a)/\alpha \right) da$. Subtracting the expressions for $\pi^*$ and $\pi$ yields the claimed identity. $\square$

**Proposition 1** (Monotone ascent of the joint objective by block updates). *With critics and $\lambda$ fixed, consider the block-coordinate ascent: (i) $\pi$-update: $\pi \leftarrow \pi^*$ (the Boltzmann improvement of Lemma 1), (ii) policy-encoder update: $\phi_\pi \leftarrow \phi_\pi + \eta \, \nabla_{\phi_\pi} I(Z; L)$ (or any step that does not decrease $I(Z; L)$). Then the joint objective $\mathcal{F}(\pi, \phi_\pi; \lambda)$ is non-decreasing:*

$$\Delta \mathcal{F} = \underbrace{\big[ \mathcal{J}_{\text{soft}}(\pi^*; \lambda) - \mathcal{J}_{\text{soft}}(\pi; \lambda) \big]}_{\geq 0 \text{ by Lemma 1}} + \beta \underbrace{\big[ I(Z^+; L) - I(Z; L) \big]}_{\geq 0 \text{ by encoder step}} \geq 0. \qquad (7)$$

The consequences of this design are threefold. First, the policy update ($\pi$-step) guarantees policy improvement in the soft Lagrangian sense: $\mathcal{J}_{\text{soft}}$ increases strictly at each step unless the policy is already optimal for every state. This mirrors the monotonicity arguments in standard policy iteration, ensuring consistent ascent of the return under the current regularization scheme. Second, the mutual information (MI) update is policy-specific: because $\widetilde{Q}_\lambda$ is computed using the critic's encoder—which is decoupled from the policy encoder—the MI regularizer can only affect the informativeness of the policy representation $I(Z; L)$. It cannot interfere with the critic's Bellman updates or the contraction property that underpins stable value estimation. Finally, when the $\pi$-improvement and MI-ascent steps are alternated, we obtain joint objective ascent: the global objective $\mathcal{F}$ increases monotonically, improving both the soft Lagrangian return and the safety-awareness encoded in the policy's features. This separation of roles allows the method to balance reward maximization, safety constraints, and representational informativeness in a principled manner.

### 3.3 MI ESTIMATION VIA INFONCE.

Directly computing $I(Z; L)$ is generally intractable for continuous, high-dimensional embeddings because $I(Z; L)$ in general because it requires the joint $p_{Z,L}$ in Equation 2. We therefore adopt the InfoNCE bound (Oord et al., 2018), which constructs a minibatch contrastive task using anchor–positive pairs from matching $(Z, L)$ samples and negatives from mismatched labels:

$$I(Z; L) \geq \log B - \mathcal{L}_{\text{InfoNCE}}(\phi_\pi),\tag{8}$$

where $B$ is the number of samples in the contrastive batch and $\mathcal{L}_{\text{InfoNCE}}$ is the cross-entropy loss over one positive and $B - 1$ negatives per anchor. Minimizing $\mathcal{L}_{\text{InfoNCE}}$ thus maximizes a tractable lower bound on $I(Z; L)$.

$$\mathcal{L}_{\text{InfoNCE}} = \mathbb{E}_{(z,z^+)\sim\mathcal{D}_+} \mathbb{E}_{z_{1:M}^-\sim\mathcal{D}_-} \left[ -\log \frac{\exp\left(g(z, z^+)/\tau\right)}{\exp\left(g(z, z^+)/\tau\right) + \sum_{j=1}^M \exp\left(g(z, z_j^-)/\tau\right)} \right].\tag{9}$$

where $z_i = f_{\phi_\pi}(s_i)$ are the encoded representations, $g(\cdot, \cdot)$ is a similarity function ( cosine similarity in our case), $\tau > 0$ is a temperature parameter that controls the concentration of the distribution, and $M$ is the number of negative samples per anchor.

### 3.4 FULL ALGORITHM

During environment interaction, we collect transitions and assign safety labels as described in Algorithm 1. For each transition $(s_t, a_t, r_t, s_{t+1}, c_t)$, we compute the safety label $\ell_t = \mathbb{1}[c_t = 0]$, which indicates whether the transition is safe (cost-free) or unsafe (cost-incurring). This systematic labeling process partitions our replay buffer $\mathcal{B}$ into safe and unsafe experiences:

$$\mathcal{D}_{\text{safe}} = \{(s, a, r, s', c) \in \mathcal{B} : c = 0\}\tag{10}$$

$$\mathcal{D}_{\text{unsafe}} = \{(s, a, r, s', c) \in \mathcal{B} : c > 0\}\tag{11}$$

For each training iteration, we sample a batch of anchor states $\{s_i\}_{i=1}^N$ from $\mathcal{D}_{\text{safe}}$. For each anchor $s_i$, we construct: 1. *Positive samples*: Other safe states $s_i^+ \sim \mathcal{D}_{\text{safe}}$, representing semantically similar (safe) experiences. *Negative samples*: Unsafe states $s_i^- \sim \mathcal{D}_{\text{unsafe}}$, representing semantically dissimilar (unsafe) experiences. This differs from typical contrastive approaches that rely on data augmentation to create positive pairs. Instead, we leverage the natural semantic structure provided by safety labels, which offers a more direct supervision signal for learning safety-aware representations. Partitioning the replay buffer is presented in Algorithm 2.

Our approach is *policy-agnostic* and can be integrated into any Safe Reinforcement Learning (Safe RL) algorithm. We choose Soft Actor-Critic under Lagrangian constraints (SAC-LAG). We provide the complete procedure summarized in Algorithm 3. However, the framework can be applied to any Lagrangian or projection based methods. At each environment interaction, we perform a standard Safe RL update: a minibatch is drawn from the replay buffer, the critics are trained to minimize the entropy-regularized Bellman error, the policy is updated under adaptive penalties on constraint costs using learned multipliers, and target networks are softly updated for stability. In parallel, we interleave a self-supervised representation phase at fixed intervals. During this phase, several epochs of contrastive training are run on anchor–positive–negative triplets drawn from the replay buffer. The contrastive objective pulls anchors toward their true successors and pushes them away from unrelated states, refining the encoder's ability to distinguish safe from unsafe regions. Importantly, only the policy encoder is updated in this phase, while the policy and critics remain frozen.

We update the encoder intermittently rather than at every epoch, which avoids destabilizing off-policy training, reduces gradient interference, and lowers computational overhead. This enables strong representation learning while preserving off-policy sample efficiency and training stability.

## 4 EXPERIMENTAL RESULTS

We train a Soft Actor-Critic with PID Lagrangian (SAC-LAG) agent that combines a stochastic Gaussian actor with state-conditioned standard deviation and automatic entropy tuning with double

**Algorithm 1:** COLLECT SAFETY DATA$(\mathcal{E}, \pi_\theta, \mathcal{B})$

**Input:** Env. $\mathcal{E}$, policy $\pi_\theta$, replay buffer $\mathcal{B}$
**Output:** Updated replay buffer $\mathcal{B}$
Initialize $s_0 \leftarrow$ RESET $\mathcal{E}$;
**for** $t = 0, 1, 2, \ldots$ **do**
    // action with exploration
    $a_t \leftarrow \pi_\theta(s_t) + \varepsilon_t$;
    $s_{t+1}, r_t, c_t \leftarrow \mathcal{E}.\text{step}(a_t)$;
    // transition tuple
    $\tau_t \leftarrow \langle s_t, a_t, r_t, s_{t+1}, c_t \rangle$;
    // safety labeling
    $\ell_t \leftarrow \begin{cases} 1 & \text{if } c_t = 0 \text{ (safe)} \\ 0 & \text{if } c_t > 0 \text{ (unsafe)} \end{cases}$;
    $\mathcal{B}.\text{add}(\tau_t, \ell_t)$;
    **if** *done* **then**
        $s_0 \leftarrow \mathcal{E}.\text{reset}()$;
    **else**
        $s_t \leftarrow s_{t+1}$;

**Algorithm 2:** Sample Safety-Contrastive Triplets from Replay Buffer

**Input:** Replay buffer $\mathcal{B}$, batch size $B$
**Output:** $\mathcal{X}_{\text{anchor}}, \mathcal{X}_{\text{pos}}, \mathcal{X}_{\text{neg}}$

// partition the buffer
$\mathcal{D}_{\text{safe}} \leftarrow \{(s, a, r, s', c) \in \mathcal{B} \mid c = 0\}$;
$\mathcal{D}_{\text{unsafe}} \leftarrow \{(s, a, r, s', c) \in \mathcal{B} \mid c > 0\}$;

// Sampling step
// $B$ neg., $B$ anc., $B$ pos.
**if** $|\mathcal{D}_{unsafe}| < B$ **or** $|\mathcal{D}_{safe}| < 2B$ **then**
    **return** $\emptyset, \emptyset, \emptyset$;

// unsafe states
$\mathcal{X}_{\text{neg}} \leftarrow \{s_i \sim \mathcal{U}(\mathcal{D}_{\text{unsafe}})\}_{i=1}^B$;

// positives and anchors
$\mathcal{X}_{\text{pos}} \leftarrow \{s_i^+ \sim \mathcal{U}(\mathcal{D}_{\text{safe}})\}_{i=1}^B$;
$\mathcal{X}_{\text{anchor}} \leftarrow \{s_i \sim \mathcal{U}(\mathcal{D}_{\text{safe}} \setminus \mathcal{X}_{\text{pos}})\}_{i=1}^B$;

**return** $\mathcal{X}_{\text{anchor}}, \mathcal{X}_{\text{pos}}, \mathcal{X}_{\text{neg}}$

Q-critics and soft target updates for stability. A PID Lagrangian controller Stooke et al. (2020) a per-task safety cost limit during learning, and an optional safety-aware contrastive module learns representations that separate safe from unsafe transitions. We used standard hyperparameters unless otherwise noted. The actor and critic learning rates were set to $5 \times 10^{-4}$ and $10^{-3}$, respectively, with two hidden layers of size $(128, 128)$. Automatic entropy tuning was enabled with an entropy coefficient learning rate of $3 \times 10^{-4}$ and initial value 0.005. We used Polyak averaging with $\tau = 0.05$, a 2-step return, and state-conditioned variance for the actor. Actions were squashed with tanh and clipped to the environment bounds, with double $Q$-critics and deterministic evaluation enabled. Safety-aware contrastive learning was activated by default. For the Lagrangian component, we adopted PID-controlled multipliers with gains $(K_p, K_i, K_d) = (0.05, 0.0005, 0.1)$, rescaling enabled, and the safety-constraint module switched on throughout training.

## 4.1 DATASET

We evaluate on a diverse suite of different safety-critical continuous-control environments spanning Bullet Safety Gym (Gronauer, 2022) and Safety Gymnasium (Ji et al., 2023). Each environment provides both a task reward and a scalar safety cost; the agent must maximize reward while keeping cumulative cost below a fixed limit, thereby capturing the fundamental trade-off between performance and safety in real-world control. The suite includes environments with varying morphology (e.g., car, drone, ball, point, and humanoid agents), different hazard layouts (e.g., pits, walls, moving obstacles), and multiple difficulty levels. This broad coverage enables a systematic assessment of generalization and robustness, allowing us to benchmark how algorithms handle both high-reward pursuit and strict adherence to safety limits under diverse and challenging conditions.

## 4.2 EVALUATION PROTOCOL AND SETUP

To evaluate the trained policies, we adopt a standardized protocol across all environments. Each agent is trained for a fixed budget of interaction steps, after which we periodically pause training and perform evaluation rollouts without exploration noise. Unless otherwise stated, evaluation episodes are run in deterministic mode, where the policy outputs the mean of the learned Gaussian distribution. To ensure fair comparison, all baselines and our method share identical network architectures, optimizers, and training budgets. Statistical significance of performance differences is assessed using paired $t$-tests across evaluation rollouts, with 95% confidence intervals reported where appropriate.

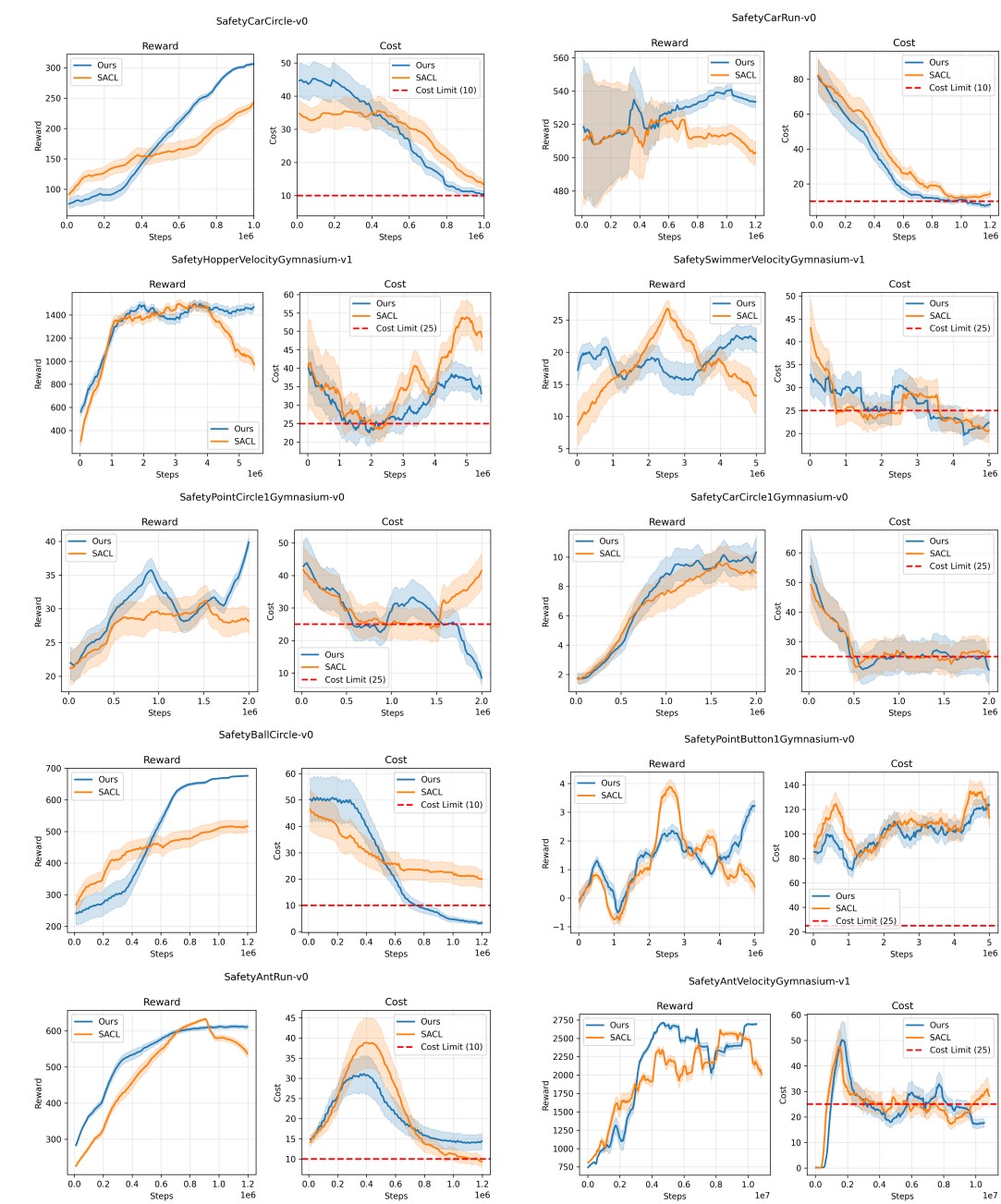

Figure 2: Training curves comparing S3RL and SAC-LAG across representative environments. S3RL achieves faster reward acquisition and smoother cost trajectories, showing improved sample efficiency and stability.

### 4.3 DISCUSSION

**Training Dynamics.** Figure 2 compares the learning curves of our method and the SAC-LAG baseline across representative environments. We observe that our approach consistently acceler- ates reward acquisition, with steeper learning curves in tasks such as `SafetyCarRun-v0` and `SafetyBallRun-v0`, where near-optimal performance is achieved with fewer interactions. In hazard-dense settings such as `SafetyDroneCircle-v0`, our method maintains smoother and

| Environment | Reward ↑ | | Cost ↓ | | Length |
|---|---|---|---|---|---|
| | S3RL | SAC-LAG | S3RL | SAC-LAG | |
| SafetyCarRun-v0 | **548.10 ± 0.00** | 542.02 ± 0.00 | 0.00 ± 0.00 | 0.00 ± 0.00 | 200.00 ± 0.00 |
| SafetyBallRun-v0 | **822.57 ± 311.74** | -411.08 ± 501.20 | 62.18 ± 33.41 | 80.02 ± 11.30 | 100.00 ± 0.00 |
| SafetyBallCircle-v0 | 635.46 ± 36.92 | **656.61 ± 39.35** | 18.26 ± 21.76 | 28.50 ± 18.69 | 200.00 ± 0.00 |
| SafetyCarCircle-v0 | **332.13 ± 0.00** | 330.08 ± 0.00 | 25.60 ± 7.10 | **5.20 ± 3.00** | 300.00 ± 0.00 |
| SafetyDroneRun-v0 | **325.53 ± 20.53** | 288.21 ± 106.89 | **15.78 ± 19.33** | 59.62 ± 30.66 | 200.00 ± 0.00 |
| SafetyAntRun-v0 | 592.11 ± 61.73 | **677.13 ± 19.00** | 13.06 ± 18.49 | **10.60 ± 5.22** | 200.00 ± 0.00 |
| SafetyAntCircle-v0 | **20.90 ± 20.48** | 13.28 ± 10.52 | 18.28 ± 25.64 | **9.10 ± 9.31** | 201.34 ± 94.28 |
| SafetyPointCircle1Gymnasium-v0 | **37.54 ± 4.44** | 22.15 ± 12.79 | **5.17 ± 5.61** | 32.65 ± 54.44 | 500.00 ± 0.00 |
| SafetyCarCircle1Gymnasium-v0 | **5.01 ± 4.69** | 7.97 ± 3.32 | **3.70 ± 3.83** | 19.23 ± 18.29 | 500.00 ± 0.00 |
| SafetyPointGoal1Gymnasium-v0 | **1.47 ± 4.11** | 0.77 ± 1.38 | **94.40 ± 55.97** | 103.25 ± 71.75 | 1000.00 ± 0.00 |
| SafetyHalfCheetahVelocityGymnasium-v1 | 2828.95 ± 50.71 | **2880.26 ± 78.87** | 1.70 ± 4.43 | **0.30 ± 1.33** | 1000.00 ± 0.00 |
| SafetyHopperVelocityGymnasium-v1 | **1163.25 ± 459.28** | 955.52 ± 401.01 | 22.32 ± 10.20 | **11.40 ± 0.00** | 699.54 ± 265.38 |
| SafetySwimmerVelocityGymnasium-v1 | **6.27 ± 17.07** | -4.32 ± 0.21 | 15.78 ± 13.94 | **0.20 ± 0.20** | 1000.00 ± 0.00 |
| SafetyWalker2dVelocityGymnasium-v1 | **3088.62 ± 94.23** | 2626.57 ± 548.82 | 51.90 ± 60.36 | **19.35 ± 1.45** | 974.65 ± 31.11 |
| SafetyAntVelocityGymnasium-v1 | **2474.75 ± 206.50** | 1994.91 ± 234.10 | 17.68 ± 18.18 | 109.20 ± 0.00 | 992.92 ± 14.16 |
| SafetyHumanoidVelocityGymnasium-v1 | **5673.13 ± 465.76** | 5028.21 ± 101.00 | 0.00 ± 0.00 | 0.00 ± 0.00 | 991.16 ± 17.68 |

Table 1: Inference performance across Bullet Safety Gym and Safety Gymnasium tasks. We compare S3RL (ours) with the SAC-LAG baseline in terms of episodic reward (higher is better), cumulative safety cost (lower is better), and average episode length. Overall, S3RL achieves consistently higher rewards in most environments while maintaining comparable or lower safety costs, highlighting the benefit of safety-aware representation learning.

lower-variance cost trajectories, while SAC-LAG exhibits larger spikes in early training. In some tasks, such as `SafetyCarCircle-v0`, the contrastive module yields higher rewards at the expense of slightly higher costs, reflecting a trade-off between task mastery and conservative safety. Overall, the curves demonstrate that safety-aware representations improve both sample efficiency and stability, producing more consistent outcomes across seeds compared to the baseline.

**Test Dynamics.** We evaluate the inference performance of 10 evaluation episodes across a diverse suite of safety-critical continuous-control environments. Table 1 reports the mean and standard deviation of the return (Reward) and safety violations (Cost), along with average episode length. Overall, S3RL consistently achieves higher task rewards in most environments, while maintaining comparable or lower safety costs in several cases. For example, in `CarRun` and `DroneRun`, S3RL outperforms SAC-LAG both in terms of reward and cost, indicating more efficient task completion under safety constraints. Similarly, on locomotion benchmarks such

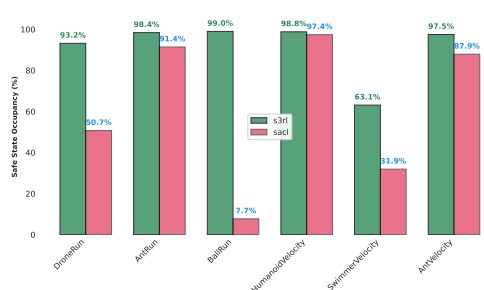

Figure 3: Safe state occupancy (%) across representative environments, comparing S3RL with SAC-LAG. Higher values indicate that the policy spends more time in constraint-satisfying regions of the state space.

as `Walker2dVelocity` and `AntVelocity`, S3RL achieves substantial gains in reward (+462.05 and +479.84, respectively).

**Insights.** The results also reveal environments where SAC-LAG exhibits an advantage. For instance, in `SafetyAntRun-v0` and `SafetyHalfCheetah`, SAC-LAG attains higher average rewards with slightly reduced safety costs, suggesting that its conservative constraint-handling may be more effective in these settings. To quantify these differences, we conducted paired statistical tests over the 10 evaluation episodes. At a significance level of $\alpha = 0.05$, the observed improvements of SAC-LAG in these two environments are statistically distinguishable, with $p < 0.05$ in both cases, and the 95% confidence intervals exclude zero. Nonetheless, the effect sizes are relatively small compared to the large and statistically significant gains of S3RL across the majority of tasks, where $p < 0.01$ confirms that the observed reward improvements are unlikely to be due to random

variation. This suggests that while SAC-LAG can provide marginal benefits in select environments, the consistent superiority of S3RL across the broader benchmark remains robust under rigorous statistical testing. Additionally, S3RL demonstrates robustness in sparse-reward environments such as `SafetyPointGoal1Gymnasium-v0`, where it achieves nearly double the reward of SAC-LAG, despite both methods incurring relatively high costs. This highlights the benefit of incorporating self-supervised representation learning in guiding exploration under safety-critical dynamics.

Figure 3 safe state occupancy (%) and absolute improvements (percentage points, pp). We observe large margins on hazard-dense tasks such as `BallRun` (+91.3 pp) and `DroneRun` (+42.5 pp), indicating that the policy quickly concentrates its visitation on feasible regions. Even on velocity-control domains where SAC-LAG is competitive, S3RL retains a consistent edge (e.g., `AntVelocity` +9.6 pp; `HumanoidVelocity` +1.4 pp). Overall, these occupancy gains align with our learning-curve trends: by shaping features to make safety-relevant structure more separable, the policy steers probability mass in the discounted occupancy measure $d_\pi(s, a)$ toward productive, low-violation regions, thereby reducing uninformative (unsafe) exploration and improving reward maximization efficiency.

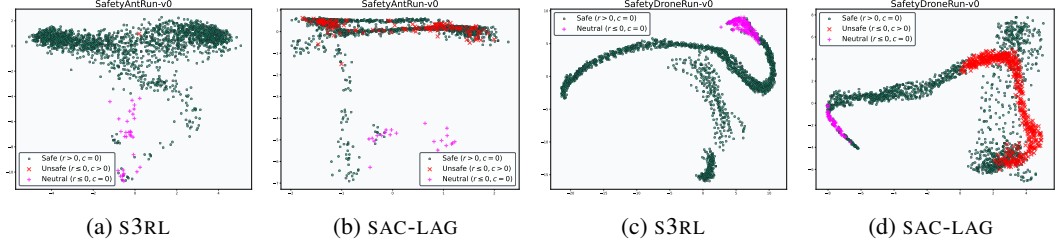

|                 (a) S3RL                  |                 (b) SAC-LAG                  |                 (c) S3RL                  |                 (d) SAC-LAG                  |

Figure 4: **State embedding visualizations using t-SNE.** Illustrating the state occupancy measure for two representative environments: `SafetyAntRun-v0` (left pair) and `SafetyDroneRun-v0` (right pair). Compared to the baseline, the S3RL embeddings exhibit clearer separation of safe and unsafe state clusters, with more compact and structured representations.

## 5 LIMITATIONS

While our approach shows consistent improvements across diverse benchmarks, several limitations remain. First, the learned safety-aware representations are derived from replay buffer partitions based on observed costs. This mechanism implicitly assumes that past unsafe experiences are sufficiently representative of the environment's hazards. In tasks where hazards are rare or highly context-dependent, the encoder may fail to capture critical risk boundaries. Second, our method focuses on binary safe/unsafe labels, which simplifies supervision but overlooks the graded nature of many real-world risks (e.g., proximity to hazards or cumulative stress on physical systems). Extending the framework to incorporate richer safety signals remains an open challenge. Third, the contrastive updates are scheduled periodically to preserve stability, yet this design introduces a trade-off: more frequent updates can destabilize training, whereas sparse updates may underutilize representation learning. Understanding this trade-off more systematically is left for future work.

## 6 CONCLUSION

Our work demonstrates that explicitly encoding safety-awareness into state representations can significantly improve both reward maximization and safety compliance in reinforcement learning. By shaping the latent space with safety labels and coupling representation learning with policy optimization, the proposed framework provides a principled way to alleviate the representation bottleneck in Safe RL. We identify a fundamental gap in Safe RL: the lack of explicit safety-aware representations. We address this gap by introducing a safety-critical inductive bias that integrates representation learning into policy optimization, leading to improved sample efficiency and more robust safety compliance. While further research is needed to explore extensions to diverse domains, scaling strategies, and more general safety signals, we believe this work takes a step towards bridging representation learning and safe decision-making in reinforcement learning.

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

## A  RELATED WORK

**Safe RL as Constrained Decision Making.**  Safe reinforcement learning is typically cast as a Constrained Markov Decision Process (CMDP) (Altman, 1999), where the agent maximizes return subject to discounted cost budgets. This formulation separates *feasibility* (constraint satisfaction) from *optimality* (reward maximization) and motivates evaluation over feasible policy sets and budgeted costs (Gu et al., 2022; Hayes et al., 2022).

**Mechanisms for Enforcing Safety.**  Three lines of work dominate. (i) *Primal–dual/Lagrangian* methods relax constraints via multipliers and alternate policy updates with dual adjustments. Constrained Policy Optimization uses trust regions and surrogate constraints (Joshua Achiam & Abbeel, 2017); PID-Lagrangian stabilizes dual dynamics for off-policy SAC (Stooke et al., 2020); recent off-policy primal–dual variants refine critic updates and improve sample-efficiency (Wu et al., 2024); variational approaches (CVPO) leverage distributional parameterizations for constrained optimization (Liu et al., 2022). (ii) *Projection/shielding* modifies actions at execution time to remain in a locally feasible set, reducing violations at the cost of potential bias and overhead (Dalal et al., 2018). (iii) *Risk-aware/distributional* methods account for tail risks and safety margins; safe distributional RL models constraints over return distributions to enable richer specifications and robust trade-offs (Zhang & Weng, 2021). Across these families, surveys emphasize reporting both reward and cumulative cost under fixed limits, across multiple seeds, with clear feasibility criteria (Gu et al., 2022).

**Representation Learning for Control.**  Orthogonal to safety mechanisms, representation learning improves data efficiency and stability in RL. Contrastive Predictive Coding (CPC) frames representation learning as future prediction with the InfoNCE objective (Oord et al., 2018); CURL applies contrastive learning to image-based control (Laskin et al., 2020). To mitigate non-stationarity from joint updates, Stooke et al. (2021) decouple representation learning from policy optimization. Eysenbach et al. (2022) connect contrastive learning to goal-conditioned value learning via discounted occupancy measures, suggesting that contrastive scores can align with control-relevant structure. These results collectively motivate *safety-aware* encoders that organize state space around hazard structure to sharpen policy gradients under constraints.

**InfoNCE: Practice and Theory.**  The InfoNCE loss underpins many state-of-the-art contrastive methods. In vision, SimCLR shows the role of augmentation, batch size, and temperature (Chen et al., 2020); Supervised Contrastive Learning leverages labels to define positives (Khosla et al., 2020); bootstrap and clustering variants reduce reliance on explicit negatives (BYOL (Grill et al., 2020), SwAV (Caron et al., 2020)); redundancy- and variance-regularized objectives temper collapse (Barlow Twins (Zbontar et al., 2021), VICReg (Bardes et al., 2022)). Theory analyzes when contrastive learning recovers discriminative features (Arora et al., 2019; Saunshi et al., 2019), clarifies mutual-information bounds and estimator trade-offs (Poole et al., 2019; Tschannen et al., 2020), and studies the role of hard negatives and temperature (Robinson et al., 2021). In RL, InfoNCE-style objectives improve sample efficiency (CURL (Laskin et al., 2020)), stabilize off-policy training when decoupled (ATC (Stooke et al., 2021)), and enhance prediction of future latent features (SPR (Schwarzer et al., 2021)).

**Our Scope and Positioning.**  Prior safe RL focuses on *how* to enforce constraints (dual control, projection, risk) but largely assumes a shared encoder that must simultaneously learn safety boundaries and maximize reward—often via trial-and-error signals. Concurrently, contrastive methods show that encoders can be shaped to reflect control-relevant structure. We bridge these threads by targeting the *state-representation bottleneck* in safe RL: we decouple a safety-aware InfoNCE/MI regularizer applied to the *policy* encoder from the *critic* encoder, preserving off-policy stability while explicitly injecting safety-discriminative geometry into policy features. This design complements primal–dual SAC-style updates (Stooke et al., 2020) and aligns with evidence that decoupling representation learning mitigates non-stationarity (Stooke et al., 2021), while leveraging contrastive objectives that connect to occupancy-aware value learning (Eysenbach et al., 2022). Our experiments adopt established safety benchmarks and protocols (Ray et al., 2019; Gu et al., 2022), emphasizing both reward gains and adherence to cost budgets.

## B   METHODOLOGY

---

**Algorithm 3:** TRAIN-S3RL-WITH-SAC-LAGRANGIAN

---

**Input**   : Replay buffer $\mathcal{B}$ with labeled tuples $(s, a, r, s', d, c, \ell)$ from Algorithm 1
COLLECT-SAFETY-DATA; policy $\pi_\theta$; critics $Q_\psi^{(i)}$ ($i{=}0$ reward, $i{\geq}1$ costs); policy
encoder $f_\phi$ (online) and optional target encoder $f_{\bar{\phi}}$; duals $\lambda$

**Output** : Updated $\theta, \psi, \phi$ (and $\bar{\phi}$), optionally $\lambda$

**Init:** Adam optimizers for $\theta, \psi, \phi$; target critics $\psi' \leftarrow \psi$.
**Schedule:** step counter $t \leftarrow 0$; encoder update interval $T_{\text{enc}}$; Encoder update epochs $E_{\text{enc}}$; Batch to
update encoder $B_{\text{enc}}$.

**while** *not converged* **do**

   // (1) Off-policy RL update
   Sample minibatch $\mathcal{D} \sim \mathcal{B}$ of size $B_{\text{RL}}$.
   For each $(s, a, r, s', d, c, \ell) \in \mathcal{D}$, draw $a' \sim \pi_\theta(\cdot \mid s')$.
   **for** $i \leftarrow 0$ **to** $N_{\text{critics}} - 1$ **do**

      **if** $i = 0$ **then**

$$y^{(0)} \leftarrow r + \gamma(1-d)\big[\min_j Q_{\psi'}^{(j)}(s', a') - \alpha \log \pi_\theta(a' \mid s')\big]$$

      **else**

$$y^{(i)} \leftarrow c_i + \gamma(1-d)\big[\min_j Q_{\psi'}^{(j)}(s', a') - \alpha \log \pi_\theta(a' \mid s')\big]$$

$$\mathcal{L}_Q^{(i)} \leftarrow \mathbb{E}_{\mathcal{D}}\Big[(Q_\psi^{(i)}(s,a) - y^{(i)})^2\Big]$$

   $\mathcal{L}_{\text{critic}} \leftarrow \sum_{i=0}^{N_{\text{critics}}-1} \mathcal{L}_Q^{(i)}$
   // Shaped policy loss (reward + safety)
   $\mathcal{L}_{\text{reward}} \leftarrow \mathbb{E}_{s\sim\mathcal{D},\, a\sim\pi_\theta}\big[\alpha \log \pi_\theta(a \mid s) - Q_\psi^{(0)}(s,a)\big]$
   $\mathcal{L}_{\text{safety}} \leftarrow \sum_{i=1}^{N_{\text{critics}}-1} \lambda_i \cdot \mathbb{E}_{s,a}\big[Q_\psi^{(i)}(s,a)\big]$
   $\mathcal{L}_{\text{policy}} \leftarrow \rho\big(\mathcal{L}_{\text{reward}} + \mathcal{L}_{\text{safety}}\big)$
   // Gradient steps & target update
   $\theta \leftarrow \theta - \eta_\theta \nabla_\theta \mathcal{L}_{\text{policy}}$
   $\psi \leftarrow \psi - \eta_\psi \nabla_\psi \mathcal{L}_{\text{critic}}$
   $\psi' \leftarrow \tau \psi + (1-\tau) \psi'$
   // Optional: PID duals (per cost $i$)
   **if** *using PID* **then**

      $e_i(t) \leftarrow \widehat{J}_{c_i} - \bar{c}_i; \quad \lambda_i \leftarrow \big[\lambda_i + K_P e_i(t) + K_I\sum_{k\leq t}e_i(k) + K_D\big(e_i(t) - e_i(t-1)\big)\big]_+$

   // (2) Periodic safety-aware SSL on policy encoder
   $t \leftarrow t + 1$
   **if** $t \bmod T_{\text{enc}} = 0$ **then**

      **for** $e \leftarrow 1$ **to** $E_{\text{enc}}$ **do**

         // Sample contrastive samples from replay buffer using
            Algorithm 2
         // Compute INFONCE loss using Algorithm 4.
         $(\mathcal{X}_a, \mathcal{X}_p, \mathcal{X}_n) \leftarrow$ SAMPLE-CONTRASTIVE-TRIPLETS$(\mathcal{B}, B_{\text{SSL}})$
         **if** $\mathcal{X}_a \neq \emptyset$ **then**

            $\mathcal{L}_{\text{InfoNCE}} \leftarrow$
            COMPUTE-INFONCE-LOSS$(X_a, \mathcal{X}_p, \mathcal{X}_n; f_\phi, f_{\bar{\phi}})$
            $\phi \leftarrow \phi - \eta_\phi \nabla_\phi \mathcal{L}_{\text{InfoNCE}}$

---

**Algorithm 4:** Compute InfoNCE Loss

**Input:** $\mathcal{X}_a, \mathcal{X}_p, \mathcal{X}_n$ (triplet batches), encoder $f_{\phi_\pi}$
**Output:** $\mathcal{L}_{\text{InfoNCE}}$
$Z_a \leftarrow f_{\phi_\pi}(\mathcal{X}_a) \in \mathbb{R}^{B \times d}$;
$Z_p \leftarrow f_{\phi_\pi}(\mathcal{X}_p) \in \mathbb{R}^{B \times d}$;
$Z_n \leftarrow f_{\phi_\pi}(\mathcal{X}_n) \in \mathbb{R}^{B \times d}$;
$\tau \leftarrow 0.7$;
$S_{\text{pos}} \leftarrow \text{diag}(Z_a Z_p^\top)/\tau \in \mathbb{R}^B$;
$S_{\text{neg}} \leftarrow Z_a Z_n^\top/\tau \in \mathbb{R}^{B \times B}$;
$L \leftarrow [S_{\text{pos}}, \ S_{\text{neg}}] \in \mathbb{R}^{B \times (1+B)}$;
$y \leftarrow \mathbf{0}_B \in \mathbb{R}^B$;
$\mathcal{L}_{\text{SSL}} \leftarrow -\frac{1}{B} \sum_{i=1}^{B} \log \left( \frac{\exp(L_{i,0})}{\sum_{j=0}^{B} \exp(L_{i,j})} \right)$;
$\mu_{\text{pos}} \leftarrow \frac{1}{B} \sum_{i=1}^{B} S_{\text{pos}}[i]$;
$\mu_{\text{neg}} \leftarrow \frac{1}{B^2} \sum_{i=1}^{B} \sum_{j=1}^{B} S_{\text{neg}}[i,j]$;
**return** $\mathcal{L}_{\text{InfoNCE}}$

## B.1 MORE EXPERIMENTS

We perform an ablation study of S3RL under both unconstrained (Non-Lagrangian) and constrained (Lagrangian) training after 100 epochs, alongside SAC, PPO, and DDPG for reference. The results show that in the unconstrained setting, S3RL achieves the highest rewards (e.g., 10 024 in HalfCheetah and 836 in DroneRun), surpassing all other methods. When Lagrangian constraints are enforced, S3RL successfully reduces costs close to zero while maintaining competitive rewards, whereas other baselines experience larger drops in performance (e.g., DDPG-LAG collapses in HalfCheetah, and SAC-LAG incurs much higher cost in DroneRun). These observations confirm that S3RL adapts more effectively than standard off-policy and on-policy methods when switching between unconstrained and constrained regimes.

Table 2: Ablation study on HalfCheetah-v1 and DroneRun-v0. All agents are trained for 100 epochs, and the reported values correspond to policy evaluation performed after training. Results are separated into Lagrangian-constrained (top) and unconstrained (bottom) settings.

| Task | Model | Reward | Cost |
|---|---|---|---|
| Half Cheetah-v1 (Lag) | S3RL-LAG | $9880.40 \pm 68.88$ | $15.87 \pm 5.66$ |
| | SAC-LAG | $9452.30 \pm 1440.87$ | $20.08 \pm 7.60$ |
| | PPO-LAG | $1930.68 \pm 272.53$ | $129.74 \pm 72.67$ |
| | DDPG-LAG | $2855.66 \pm 38.55$ | $8.97 \pm 11.81$ |
| Half Cheetah-v1 (Non-Lag) | S3RL | $10\,024.07 \pm 116.46$ | $978.15 \pm 2.22$ |
| | SAC | $9\,491.12 \pm 121.89$ | $977.85 \pm 3.13$ |
| | PPO | $2535.46 \pm 66.76$ | $412.00 \pm 30.88$ |
| | DDPG | $8940.81 \pm 1298.56$ | $956.55 \pm 35.31$ |
| Drone Run-v0 (Lag) | S3RL-LAG | $287.88 \pm 7.86$ | $0.15 \pm 0.65$ |
| | SAC-LAG | $174.26 \pm 17.88$ | $39.25 \pm 1.92$ |
| | PPO-LAG | $55.43 \pm 257.29$ | $87.85 \pm 44.87$ |
| | DDPG-LAG | $384.54 \pm 6.71$ | $55.30 \pm 3.32$ |
| Drone Run-v0 (Non-Lag) | S3RL | $836.27 \pm 18.74$ | $155.55 \pm 0.73$ |
| | SAC | $803.77 \pm 22.57$ | $152.15 \pm 0.57$ |
| | PPO | $729.77 \pm 17.93$ | $150.00 \pm 0.00$ |
| | DDPG | $738.02 \pm 67.75$ | $153.50 \pm 8.43$ |

| Name | Description | Default |
|------|-------------|---------|
| **General** | | |
| task | Environment ID | SafetyDroneRun-v0 |
| cost_limit | Per-task safety cost limit | 10 |
| device | Compute device | cuda |
| thread | CPU threads when using CPU | 4 |
| seed | Random seed | 10,20,30,40,50 |
| **SAC (policy/value)** | | |
| actor_lr | Actor learning rate | 5e-4 |
| critic_lr | Critic learning rate | 1e-3 |
| hidden_sizes | MLP hidden layer sizes | (128, 128) |
| auto_alpha | Automatic entropy tuning | True |
| alpha_lr | Entropy coefficient LR | 3e-4 |
| alpha | Initial entropy coefficient | 0.005 |
| tau | Target network Polyak factor | 0.05 |
| n_step | N-step return | 2 |
| conditioned_sigma | State-conditioned std for actor | True |
| unbounded | Disable tanh squashing | False |
| last_layer_scale | Scale last layer init | False |
| gamma | Discount factor | 0.99 |
| deterministic_eval | Deterministic evaluation | True |
| action_scaling | Scale actions to bounds | True |
| action_bound_method | Action bound handling | clip |
| use_double_critic | Use double Q for critics | True |
| use_contrastive | Enable safety-aware contrastive learning | True |
| **Lagrangian (safety constraint)** | | |
| lagrangian_pid | $(K_p, K_i, K_d)$ gains | (0.05, 0.0005, 0.1) |
| rescaling | Rescale constraint penalty | True |
| **Collection / Training** | | |
| epoch | Training epochs | 100 |
| enc_training_epoch | Number of epochs to train encoder | 50 |
| enc_update_steps | Steps to update encoder | 1000 |
| episode_per_collect | Episodes collected per cycle | 20 |
| step_per_epoch | Env steps per epoch | 10000 |
| update_per_step | Gradient updates per env step | 0.2 |
| buffer_size | Replay buffer capacity | 100000 |
| training_num | Parallel training envs | 10 |
| testing_num | Parallel testing envs | 2 |
| batch_size | SGD mini-batch size | 256 |
| reward_threshold | Early-stop reward threshold | 10000 |
| save_interval | Epochs between checkpoints | 4 |

Table 3: Hyperparameters for training and evaluating S3RL.

## B.2 LLM USAGE

When preparing this manuscript, we utilized a Large Language Model (LLM) to assist with various aspects of the writing and research process. The LLM was employed for several key tasks:

- **Grammar and Language Polishing:** The LLM helped improve sentence structure, grammar, and overall readability of the manuscript, ensuring clear and professional academic writing throughout the paper.

- **Formatting Consistency:** We used the LLM to check and maintain consistent formatting across sections, including proper citation formatting, equation numbering, and LaTeX structure.

- **Technical Writing Assistance:** The LLM provided support in crafting clear explanations of technical concepts, improving the clarity of mathematical formulations, and ensuring consistent terminology throughout the paper.

It is important to note that all technical content, experimental results, mathematical derivations, and core research contributions remain entirely our own work.

