# OpenReview forum: "Safety-Aware Reinforcement Learning via Contrastive State Representations"
_ICLR.cc/2026/Conference — ICLR 2026 Conference Withdrawn Submission_

### Official Review · Reviewer_wH3G · 2025-10-27

**Soundness:** 2
**Presentation:** 3
**Contribution:** 2
**Rating:** 2
**Confidence:** 4

**Summary:**

This paper aim at tackling the challenge of safe RL (finding a policy which optimizes a reward while satisfying constraints). To do so, an additional contrastive training objective is proposed on the policy over the current SAC algorithm : its latent space is encouraged to capture mutual information with the safety label of a state, which was observed in the replay buffer. The insight is that this should improve the safety-related state representation of the policy.
The approach is tested on many environments from Safety Gymnasium and Bullet Safety Gym.

**Strengths:**

The paper’s main insight, i.e. that representation learning is an important challenge in Safe RL, is relevant. The idea of adding a safety-critical inductive bias leveraging safety labels is interesting. The contrastive learning approach is novel in this context, and it is sensible. The paper showcases extensive experiments with many environments.

**Weaknesses:**

The main weakness of this paper is that the experimental results do not support the claims. Although the approach is sensible, the presented performance in figure 2 and table 1 are really unconvincing compared to the baselines.
On the plots, there is so much variability in the curves, which pass one over each other several times during training, that it seems the experiments were stopped just at the right moment for S3RL to be better than SAC-LAG. A clear example is SafetySwimmerVelocityGymnasium-v1 where clearly SAC-LAG learns faster and better but suffers from catastrophic forgetting after having reached the highest score with a cost under the limit. SafetyHopperVelocityGymnasium-v1 has the same thing, where stopping the experiment at 2e6 steps would have given very similar (and good) performances for both algorithms. This generalizes to most plots.
The table results also show several issues, especially given the 95 confidence intervals. In most cases, it is impossible to say that one method is better than the other. In the text, SafetyPointGoal1Gymnasium-v0 is underlined as the rewards almost doubles... but it goes from 0.77 +- 1.38 to 1.47 +- 4.11, and similarly for the costs, which is absolutely not significative. This is not a unique case: most comparisons have the same issue.

In addition, the authors declare having used a p-test methodology to ensure their results are significative. This is a good reflex: it should be more used in RL papers. However, this specific methodology is not described and it is difficult to know, for example, how many runs were used for the results (especially problematic when the results show a confidence interval of 0.00...)

Another important weakness is the confusing and inconsistency in the notation and the results presentation. There are several undefined elements introduced in the equations, and the text discusses results that are not presented in the figures. There is even a figure (4) which is not referred to in the main text.

**Questions:**

* The whole paper is about enforcing mutual information between safety and the policy's latent representation $f_{\phi_\pi}$. Which latent representation are we talking about? Is it a hidden layer (which one, how is it chosen)? Or an additional pre-encoding step?
* What is the induced occupancy introduced in equation (1)? Is it conditioned on (s,a) as in the equation  or on (z,a) as described in the text?
* Figure 3 - what does 7.7% of safe-state occupancy mean here for BallRun with SACL? It does not make sense to me - 92% of the time, the agent is in unsafe states,  compared to 1% only for S3RL? How does (92% unsafe compared to 1% unsafe) lead to the cost scores reported in Table 1 : (80 compared to 62) ? This seem really incoherent.
* Related work: Mani et al. (2025) Safety Representations for Safer Policy Learning in ICLR have a similar idea of enforcing risk-based representations for Safe RL through an inductive bias. Could it be an useful piece of work to build upon and a potential baseline?

Comments

* Why do we talk about the encoder in preliminaries (section 2) about safe-RL? It seems to pertain to the method.
* Shouldn't the Mutual information part 3.1 be in the preliminaries?

---

### Official Review · Reviewer_XtH2 · 2025-11-01

**Soundness:** 2
**Presentation:** 2
**Contribution:** 2
**Rating:** 2
**Confidence:** 4

**Summary:**

The paper proposes to tackle the Safe RL from representation learning perspective and propose a contrastive safe state representation. The idea is to learn which states are safe and which are not through a contrastive loss. The definition of safe vs unsafe states is guided by the safety cost (zero vs positive) incurred at that state.

**Strengths:**

* It’s an interesting idea to learn the safety cost representation at a specific state and a contrastive approach is novel to my best knowledge
* The presentation of the paper is good and the contribution is well explained
* Visualisation in Figure 4 adds value to the paper and provide insights into the results

**Weaknesses:**

* The authors say that the approach can be used with any algorithm, but explore only one - PID Lagrangian with SAC. This claim should be supported by some evidence
* The results are not conclusive in my opinion.
     * Many training curves do not show good convergence properties, and only a couple show good separation between the curves and somewhat ok convergence
     * The presentation of results in Table 1 is incomplete without the cost limits. Do we actually solve the safety problem in any of these cases? (see questions)
* The formulation is slightly confusing: why `c = 0` is safe and `c > 0` is unsafe? In some environments incurring safety cost is part of the game as long as we satisfy the constraints at the end of the episode. In my point of view, the states are unsafe when we violate the constraint i.e., the accumulated cost is higher than the cost limit.

**Questions:**

* How are the costs computed in the considered environments? It can be relevant to the definition of safe (c=0) and unsafe (c>0) states.
* In Table 1 the statement that lower costs are better is not correct. The costs have to be lower than the cost limits.
* What are the cost limits in the environments in Table 1?

---

### Official Review · Reviewer_nsZD · 2025-11-01

**Soundness:** 1
**Presentation:** 2
**Contribution:** 1
**Rating:** 2
**Confidence:** 5

**Summary:**

The paper argues that learning state representations from scratch alongside the policy can be challenging and may result in an extended exploration phase, which is undesirable in the context of safe reinforcement learning. To address this, it proposes a method for learning safety-aware state representations by maximizing the mutual information between state embeddings and corresponding safety labels using contrastive learning. The authors further contend that incorporating these safety-aware representations enhances the performance of the underlying algorithms, improving both efficiency and safety.

**Strengths:**

1. The paper addresses the important challenge of safe exploration, where it is crucial to balance task performance with adherence to safety constraints during learning.

2. It correctly identifies that poor state representations can hinder safe exploration and proposes learning safety-aware representations. These representations encode safety-relevant information into the state embeddings by maximizing the mutual information between the embeddings and the safety labels.

**Weaknesses:**

**[W1] Missing Key References and Baselines**

The importance of learning state representations that encode safety information is not novel and has been explored in prior work, such as [1]. However, the paper does not cite this existing literature and fails to compare against this relevant baseline. Including such comparisons would strengthen the contribution of the paper.

**[W2] Unconvincing Empirical Results**

The paper claims that the proposed method improves both safety and efficiency, but the empirical evidence does not fully support this claim. This is partly due to two factors:

a) **Limited and potentially weak baseline**: The evaluation considers only SAC-Lag as a baseline. SAC-Lag is known to be unstable due to bootstrapping errors in off-policy RL. Comparisons with more stable and widely used safe RL algorithms, such as CPO [2], PID-Lag [3], and CVPO [4], would provide a clearer picture of the method’s effectiveness.

b) **Unclear safety-performance tradeoff**: From the presented plots, it is difficult to determine whether the proposed method consistently improves performance. In some environments, it achieves higher task performance at the cost of safety, while in others, safety improves at the cost of performance. A more informative evaluation would show the safety-efficiency tradeoff across varying levels of constraint enforcement during training, for example via a Pareto frontier.

References:

[1] Mani et al., Safety Representations for Safer Policy Learning, ICLR 2025

[2] Achiam et al., Constrained Policy Optimization, ICML 2017

[3] Stooke et al., Responsive Safety in Reinforcement Learning by PID Lagrangian Methods, ICML 2020

[4] Liu et al., Constrained Variational Policy Optimization for Safe Reinforcement Learning, ICML 2022

**Questions:**

1. How does the proposed method for learning state representations improve upon [1]?

2. Does the safety-aware state representation learned by the proposed approach lead to a better safety-performance tradeoff compared to existing baselines? It would be helpful if the authors could present a Pareto frontier to illustrate this tradeoff.

Minor points:
The paper claims that the proposed method is self-supervised; however, it appears to rely on safety labels derived from the environment or the cost function. In that case, the supervision is not entirely self-generated. The authors should clarify in what sense the approach qualifies as self-supervised.

[1] Mani et al. Safety Representations for Safer Policy Learning (ICLR 2025)

---

### Official Review · Reviewer_k69T · 2025-11-04

**Soundness:** 2
**Presentation:** 1
**Contribution:** 2
**Rating:** 2
**Confidence:** 4

**Summary:**

The paper tackles a representation bottleneck in safe RL: state representation may not capture safety-relevant structure. S3RL adds a safety-aware encoder trained with InfoNCE to maximize mutual information between embeddings and binary safety labels (safe vs unsafe), while the control algorithm is a standard Lagrangian SAC. Theory shows soft policy improvement and monotone ascent of a joint objective when critics and the dual variable are fixed and experiments show faster convergence to safe, rewarding policy in some tasks.

**Strengths:**

* **Clear motivation:** Making safety structure explicit in the representation is intuitive and should aid safety exploration better than vanilla state representation.
* **Modularity:** The training scheme decouples representation learning from policy learning, allowing S3RL to be plugged into SAC-Lagrangian loop.
* **Practical refinement:**: S3RL stabilizes training by doing periodic encoder updates and Fig3 shows qualitative occupancy plots (the plot doesn't make sense in some tasks, however. Please refer to weaknesses).

**Weaknesses:**

## Positioning & Clarity

* **s vs z mismatch**: The problem setup (Eq1) is in state space s, but $d^\pi(z,a)$ in Section 2 is introduced without clearly stating which parts of the algorithm use z vs s. Relatedly, critics seem to use original state features while the policy uses the MI-trained encoder; but I couldn't find these details in any of the equations.

* **Binary labels lose cost magnitude:** The labeling process lumps all positive costs into "unsafe", disregarding the cost magnitude. In the tasks experimented, this may be ok since cost is binary. But general CMDP allows for real-valued cost. Ignoring the magnitude may cause the method to fail to distinguish "very unsafe" and "slightly unsafe" transitions.

* **State-only labeling for action-dependent risk**: Costs are $c(s,a)$, yet only states are used in MI as positive and negative samples. It's unclear why state-only contrastive is sufficient when CMDP cost is state-action dependent in Section 2.

* **Trade-offs under-discussed**: While the state specification is often not designed to be informative about safety, it's usually highly relevant for task success (i.e. reward). Pushing the policy to rely on a new latent encoder can harm reward representations.

* **Unclear significance of assumptions**: Lines 142-144 state that latent representation where safety-relevant structure "is linearly separable and predictive of future violations". I'm not sure why is this assumption required for S3RL to work.

* **Theory scope vs practice**: Proposition 1's monotone ascent requires fixed critics/dual but the algorithm updates them in practice.

* **Novelty is modest**: The paper spent large amounts of text discussing prior work. For example, InfoNCE, Mutual Information, Lemma 1 (which reads as a restatement of the standard soft policy improvement theorem). There don't seem to be many novel components and novelty seems quite limited.

* **Hyperparameter $\beta$**: $\beta$ is used in policy improvement objective but I don't see its value being reported. Guidance should be provided on how this hyperparam is set or adjusted.

## Experiments & Reporting

* **Missing cost limits**: Cost limit for each task isn't listed in Table1. This makes it impossible to judge which policy is safe/unsafe. Moreover, from Fig2, there seem to be different cost limits set for different tasks (10 vs 25). Varying cost limit makes it difficult to compare the safe RL performance with the established benchmark study.

* **Small evaluation budget**: Only 10 eval episodes are used for evaluation.

* **Inconsistent result between Fig2/Fig3 and Table1**: Fig2 doesn't seem to match Table1 figure in some cases. Even factoring in deterministic evaluation mode, the discrepancy is too huge. For example:
    * CarCircle: In Fig2, SACL is still unsafe at convergence, but in Table1 it's very safe with expected cost of around 5. In line 402, it's said that S3RL yields higher reward with slightly higher cost but this is not aligned with Table 1 (S3RL exhibits similar reward but much higher cost and above the threshold 10).
    * In Fig3, S3RL shows very high safe state occupancy in DroneRun and BallRun compared to SACL. However, this is not reflected in Table1. If safe state occupancy is truly that high, we should see very low cost incurred by S3RL in these two tasks. In fact, Table1 shows that S3RL still incurs high cost in these two tasks.

* **Unsupported or conflicting claim on outperformance**: Some of the claims made in the text is unsupported due to lack of visual charts. For example:
    * It is claimed that S3RL "maintains smoother and lower variance cost trajectories" in DroneCircle. But the figure is not shown.
    * It is claimed that S3RL "outperforms in both reward and cost" in CarRun. But from Table1, the difference in reward is very small (and may be statistically insignificant, and both incur zero cost.
    * It is claimed that S3RL "achieves very good reward in WalkerVelocity". But Table1 shows that S3RL is unsafe with high cost.
    * It is claimed that S3RL "achieves double reward of SACL" in PointGoal. But Table1 shows the cost incurred for both algorithm is unsafe and very high (>90). Safe RL algorithm prioritizes cost when constraint isn't met. Comparing reward value is meaningless when safety isn't met.

* **More result discussion**: In Fig2, S3RL fails to converge to safe policy in Hopper, AntRun & PointButton, faring worse than SACL sometimes. The paper could discuss more about them.

* **No unsafe region in t-SNE**: Unsafe regions don't show up in Fig4b and 4d. This makes the result not interpretable.

## Others

* No code is provided.
* Minor typo at line 044: not not.

**Questions:**

1. Precisely which components consume s vs z? Do both critics and the dual use the same encoder as the policy?
2. Why state-only MI is used for action-dependent costs?
3. Can S3RL handle costs with varying magnitude rather than binary labels?

---

### Note · Authors · 2025-12-01

**Comment:**

Dear Reviewers and Area Chair,

After carefully reviewing your feedback, we have decided to withdraw our submission. We agree that the paper requires significant revisions and additional experimentation, which cannot be properly addressed within the short time frame.

We sincerely thank you for your time, as well as for the detailed and constructive reviews, they have been very useful in identifying important gaps in our current work. We intend to incorporate your suggestions to improve the quality of the paper for a future submission.

Thank you.

**Withdrawal Confirmation:**

I have read and agree with the venue's withdrawal policy on behalf of myself and my co-authors.